# An Investigation of the Beta Anomaly in Emerging Markets: A South African Case

**Mabekebeke Segojane and Godfrey Ndlovu *** 

School of Economics, University of Cape Town, Cape Town 7701, South Africa; sgjmab001@myuct.ac.za
* Correspondence: godfrey.ndlovu@uct.ac.za

**Abstract:** High-risk stocks tend to provide lower returns than low-risk stocks on a risk-adjusted basis. These results (referred to as the low-beta anomaly) run counter to theoretical expectations. This paper examines the beta anomaly in one of the largest emerging markets in Africa, the Johannesburg Stock Exchange (JSE). It employs both time-series and cross-sectional econometric techniques to analyze the risk–return relationship implied by the CAPM, using data that span over 5 years and 220 companies. To check for robustness, the analysis period was extended to 10 years, and we also applied the Fama–French three-factor model. The findings suggest the existence of the beta anomaly and a negatively sloped SML, indicating that beta is not the only determinant of risk in the South African stock market. We also found positive beta–idiosyncratic volatility (IVOL) correlations. However, after controlling for IVOL and the adverse effects of COVID-19 for an extended study period, the beta anomaly disappeared.

**Keywords:** beta anomaly; idiosyncratic volatility; JSE; emerging markets

## 1. Introduction and Background

The capital asset pricing model (CAPM) postulated by Sharpe (1964) and Lintner (1965), with contributions from Treynor (1961) and Mossin (1966), asserts that riskier stocks (high beta) provide higher returns than less risky stocks (low beta). Given that investors are risk-averse, they expect greater returns as compensation for the risk they bear when investing in riskier assets. The beta anomaly, therefore, is the observation that the theoretical risk–return relationship implied by the CAPM does not hold empirically. Furthermore, under the CAPM, the opportunity to increase returns without increasing risk is considered an anomaly significant enough to challenge the very foundations of efficient markets. Therefore, the persistence of contradictory empirical findings to the CAPM in the beta anomaly invites further inquiry into the anomaly. For emerging markets in particular, the beta anomaly becomes increasingly important on account of the greater structural differences, volatility, illiquidity, and information asymmetries that contribute to risk in these regions (Lim and Brooks 2010; Nishiotis 2002; Bekaert and Harvey 2002, 2003). Consequently, a better understanding of the low-beta anomaly is necessary if greater levels of investment and investor confidence in emerging markets are to be achieved.

Several theoretical postulations have been provided in the literature to explain the low-beta anomaly. From a behavioral finance perspective, explanations ranging from individual investor demand for assets with a high probability of quick short-term return, referred to as lottery returns (Bali et al. 2017; Baker et al. 2011), and highly leverage-constrained institutional investors (pension funds and mutual funds) seeking alternative avenues to through which to increase their expected returns have been proposed (Frazzini and Pedersen 2014; Bali et al. 2017). The mechanism underlying the anomaly is such that increased purchases of high-beta stocks, induced by leveraged entities or lottery investors, create upward (downward) price pressure on high (low) beta stocks, which increases (decreases) the prices of the stocks and leads to decreases (increases) in the returns of the

high (low) beta stocks (Bali et al. 2017). Other notable explanations attribute the anomaly to investor irrationality, which perpetuates the established bias of representativeness and overconfidence in the market, in addition to margin requirements (Jylhä 2018; Vapola 2019). For irrational investors, overconfidence results in the underestimation of risk, especially for high-risk firms (high-beta stocks), which results in abnormally high behavioral demands for speculative trading in those stocks, lowering their returns (Han et al. 2020).

Other explanations of the anomaly are provided within the paradigm of model mis-specification. These associate it with model mis-specification in the CAPM or unaccounted-for determinants of the risk–return relationship. According to this perspective, the risk–return relationship does not stem from market risk alone but is rather a function of additional risk factors, such as profitability, liquidity, momentum—and, more emphatically, the value and size effects. Empirically, and over the long run, it has been observed that average returns from small capitalization stocks have outperformed those from large capitalization stocks (size effect) (Basu 1977; Banz 1981). Likewise, high book-to-market stocks (value stocks) have outperformed low book-to-market stocks (growth stocks) (Reid et al. 1985; Davis 1994; Fama and French 1993). Therefore, accounting for any of these factors should eliminate the beta anomaly. However, evidence suggests that the anomaly may still hold (Barroso et al. 2016). For this paper, an in-depth analysis of the factor contributions to the beta anomaly is omitted, as it is not integral to the adopted approach. Nonetheless, while different explanations for the anomaly are provided, their insights present possible arbitrage opportunities for portfolio construction strategies that aim to exploit the workings of the anomaly.

Globally, the beta anomaly has predominately been reported in developed markets, with few articles reporting on its role in emerging markets, especially South Africa. Ghysels et al. (2016) suggest that the greater volatility of emerging markets, coupled with their asymmetric return distributions, make them ideal for investors seeking lottery returns. This results in a complication of the beta anomaly in emerging economies. This is particularly important when considering the common observation of a counterintuitive negatively sloped security market line (SML) in emerging markets, compared to a flat one or positive one in developing markets (Han et al. 2020). For a large emerging economy such as South Africa, which is the second largest economy in Africa, understanding the beta anomaly becomes even more important given the investment strategies that seek to exploit it. The "betting again beta" (BAB) strategy developed by Frazzini and Pedersen (2014) is one such strategy. Consequently, improving our empirical understanding of the anomaly is imperative to assess the applicability of the portfolio construction strategies on the Johannesburg Stock Exchange (JSE) predicated on its implications.

One study that sought to bridge the gap between the low-beta anomaly and investment on the JSE was presented by Bradfield and Oladele (2018). Their study looked at the construction and performance of low-volatility portfolios in South Africa over the period between 2006 and 2016 (Bradfield and Oladele 2018). Their findings suggest that low-volatility blended portfolios substantially outperform the All-Share Index (ALI) (Bradfield and Oladele 2018). While their study provides useful insights into portfolio construction strategies, and justification for blended low-volatility portfolios in South Africa, they did not directly test for the beta anomaly; the anomaly is instead inferred in their findings. However, testing for the anomaly is necessary to understanding its implications for investment in South Africa. This study seeks to contribute to the literature on the low-beta anomaly by testing whether the low-beta anomaly is present on the JSE, using the CAPM, and identifying possible explanations for the anomaly. Moreover, it seeks to contribute to the breakthroughs of Bradfield and Oladele (2018) in the South African environment by revisiting the implications of their recommended low-volatility portfolio construction strategies with more recent data.

The approach employed herein differs from that of Bradfield and Oladele (2018) in a number of ways. This paper includes the period after South Africa's credit was downgraded and the global COVID-19 pandemic. This paper, unlike the Bradfield and

Oladele (2018) study, includes a robust analysis of the beta anomaly. This allows the paper to decompose risk into systematic and non-systematic risk (idiosyncratic volatility (IVOL)) to test for the beta anomaly, and thus examine the role of market risk in predicting expected returns from a unique perspective. The risk decomposition is rooted in the IVOL, alpha, and beta correlations that are postulated to perpetuate the beta anomaly via a mispricing mechanism (Liu et al. 2018). The correlated factors are also associated with the IVOL anomaly, an anomaly with similar risk–return implications as the beta anomaly, but with reference to non-systematic risk as the driver of the phenomena. Thus, this study provides a unique approach to the investigation of the low-beta anomaly in South Africa through its incorporation of the IVOL, beta, and alpha relations to explain the beta anomaly, as opposed to previous studies, which primarily examined the existence of the beta anomaly as such.

The rest of this paper is organized as follows: Section 2 provides a theoretical and empirical literature review of the CAPM and low-beta anomaly, Section 3 discusses the data and methodology employed, Section 4 focuses on the preliminary analysis, Section 5 discusses the empirical results, and Section 6 summarizes the major conclusions and provides recommendations based on the study.

## 2. Literature Review

The CAPM has its foundations in modern portfolio theory, developed by Markowitz (1952). It is underpinned by the assumptions of market efficiency and investor rationality. In the standard model introduced by Sharpe (1964) and Lintner (1965), with contributions from Treynor (1961) and Mossin (1966), the relationship between risk and expected return is based entirely on systematic, undiversifiable risk. The model suggests a linear and positive relationship between the beta and the asset return, implying a positively sloped security market line (SML). Furthermore, in the standard Sharpe (1964) CAPM, the slope of the SML is equal to the risk premium, which is the difference between the expected return on the market and the expected return on the risk-free asset. The other component of risk, idiosyncratic risk, is not accounted for in the CAPM, on the assumption that it may be diversified away (Sharpe 1964). If the CAPM holds, therefore, the market beta captures all the effects of systematic risk on the expected returns of a security. Empirically, CAPM predictions have not been accurate, resulting in market anomalies that have cast doubt on the predictive capacities of the model.

Interest in the beta anomaly has soared since the first test of the CAPM by Black et al. (1972) found a flatter SML (i.e., small relationship between beta and asset return) than that implied by the CAPM. While their tests supported the linearity of the SML, their results sparked a debate over the validity of the CAPM, with decisive empirical evidence against the model remaining elusive. Early supportive explanations for the difference in SML steepness attributed it to the assumption that investors can borrow and lend on risk-free assets (Sharpe 1964). On the other hand, tests by Fama and MacBeth (1973), using a two-pass regression process, and Blume and Friend (1973), with a three-period test on the New York Stock Exchange (NYSE), confirmed the positive risk–return relationship implied by the CAPM, denying the potential existence of the beta anomaly. Despite the vicissitudes of the CAPM and its empirical tests, however, internal pressure to advance explanations of the beta anomaly have significantly increased the focus on the anomaly in the asset-pricing literature. Two prevalent approaches to explaining the anomaly are the model mis-specification and behavioral finance perspectives, with tangential explanations attributing the anomaly to deficiencies in the methodologies used to estimate betas and adequately test (statistically) for the anomaly.

One issue taken to perpetuate evidence for the beta anomaly is related to the estimation of the beta coefficients themselves. Sharpe (1964) noted that the expected returns on a given asset, or portfolio of assets, change with changing values of beta, leading to imprecise estimates. Building on Sharpe's point further, Blume (1971) revealed that the stability of betas varies from individual assets to portfolios of assets, with beta stability improving



from individual assets to portfolios of assets, and improving further with an increase in the number of assets in any given portfolio. Abdymomunov and Morley (2011) and Bos and Newbold (1984) provide results that confirm this observation, suggesting that earlier studies on the beta anomaly may have provided unreliable beta anomaly results on account of errors in the beta estimations.

Similarly, McEnally and Upton (1979) and Pettengill et al. (1995) assert that earlier CAPM studies failed to accurately ascertain the relationship between beta and returns, and consequently the beta anomaly, by not disaggregating periods of positive and negative market returns (factoring in the conditional relation between beta and returns). They argue that, given that positive markets periods yield positive beta–return relationships, while negative market periods yield negative beta–return relationships, a reliable test of the beta–return relationship should necessarily distinguish between the two periods (positive and negative market periods) if the results obtained are to adequately explain the beta anomaly (McEnally and Upton 1979; Pettengill et al. 1995). While these methodological arguments are important, they appear as tangential issues in the literature on the beta anomaly. The major explanations stem from the IVOL–beta relationship and behavioral finance explanations of the anomaly.

A closely related anomaly to the beta anomaly is the idiosyncratic volatility (IVOL) anomaly introduced by Ang et al. (2006). It presents the risk–return anomaly from the perspective of non-systematic risk, estimating risk using IVOL as a key driver of the anomaly. Regarding its connection to the beta anomaly, the IVOL perspective attributes the anomaly to the force exerted by IVOL on stock prices and the resultant effect on the beta (Liu et al. (2018). Incidentally, the beta anomaly is driven by a combination of positive beta–IVOL and negative alpha–IVOL correlations, which are postulated to arise only in overpriced (high-IVOL) stocks (Liu et al. 2018). Hence, the beta anomaly is persistent where stocks are overpriced and there is a high correlation between the beta and IVOL. The role of IVOL in the beta anomaly also brings to light an early argument that beta is not the principal measure of expected returns, highlighting the importance of non-systematic risk in explaining the expected returns on both the beta and IVOL anomalies (Ang et al. 2006).

The IVOL perspective also presents a critique of the established beta-driven explanations of the anomaly, which are presented in turn. Blitz and Van Vliet (2007), in a study on large Japanese and European stock data, using decile portfolios and the FTSE world development index, found that stocks with low volatility yield higher risk-adjusted returns compared to high-volatility stocks. They attribute their results to leveraging constraints and behavioral biases towards lotteries (stocks with a large probability of short-term returns). Black et al. (1972) and, more recently, Hwang et al. (2018), provided similar results. In a significant 2014 study, Frazzini and Pedersen (2014) attributed the beta anomaly to the leverage constraints experienced by institutional investors (pension and mutual funds), who face marginal constraints that limit their access to leverage. The mechanism underlying the anomaly is such that higher purchases of high-beta stocks induced by leverage-constrained entities, or lottery investors, create upward (downward) price pressure on high (low) beta stocks, which increases (decreases) the prices of the stock and leads to decreases (increases) in the returns on the high (low) beta stocks (Frazzini and Pedersen 2014).

Subsequently, Bali et al. (2017), using cross-sectional regressions and a univariate analysis of US data from 1963 to 2012, found that the beta anomaly disappeared when beta-sorted portfolios were naturalized to lottery demand. More recently, Jylhä (2018) and Vapola (2019) corroborated the earlier studies on the role of marginal requirements in explaining the beta anomaly. Other explanations have attributed the anomaly to investor overconfidence (Antoniou et al. 2016) or short-selling impediments (Hong and Sraer 2016). However, the major critique of the IVOL perspective on beta-driven explanations has been that the beta-driven reasons to do not sufficiently explain why investor preference would be restricted to overpriced, high-beta stocks for reasons unrelated to beta (Liu et al. 2018). Consequently, a key feature of the IVOL perspective is that the beta anomaly is a function of a mis-specification in the CAPM. One solution provided in the literature has been to

improve the CAPM, as demonstrated by the introduction of the Fama–French 3, 5, and other factor models, which have been credited with severe reductions in the beta anomaly (Karp and Van Vuuren 2019). On the other hand, behavioral finance provides an equally plausible alternative through which to explain the beta anomaly in the literature.

The behavioral finance view is generally antagonistic to the mis-specification view, yet none has managed to prevail over the other. According to behavioral finance, the beta anomaly is underpinned by investor bias and irrationality, and it is influenced as much by psychological biases as by bounded rationality and emotional factors (Barberis and Thaler 2003). Consequently, investors rely on computational shortcuts, emotional responses, choice heuristics, and other irrational methods when making decisions in an otherwise uncertain world (with inefficient markets) (Barberis and Thaler 2003). The beta anomaly is thus driven by investor attention that is disproportionately diverted to high-risk stocks on account of their visibly higher return potential, as opposed to low-risk, 'invisible' stocks (Blitz et al. 2019). The result is the perpetuation of excessive pressure (a consequence of the choice heuristic) to buy high-risk stocks and the consequent overpricing of these stocks, which leads to their lower returns (Blitz et al. 2019). Another explanation is the theory of market under- and over-reaction, which regards market inefficiencies that perpetuate the beta anomaly as functions of investor overconfidence and biased self-attribution on the part of investors (Daniel et al. 1998). Baker et al. (2011) present related explanations, including investors' representativeness bias, their preference for lottery assets, and their overconfidence as drivers of the beta anomaly.

The beta anomaly perpetuation mechanism based on overconfidence and representativeness bias operates through increases in the purchase of highly volatile stocks driven by investors' over-optimistic assessment of the prospective returns on highly volatile stocks (those presumed to be on the rise), which causes the stocks to become overpriced and, subsequently, generate lower returns (Baker et al. 2011; Blitz et al. 2019). Thus, behavioral finance explanations of the beta anomaly have proven to be both thought-provoking and informative, even under varying methodologies, in markets in both developed and developing countries. However, there is no established consensus on the best approach (between behavioral finance and mis-specification) to explaining the beta anomaly in the literature. This study henceforth discounts aspects of behavioral finance in the results and interpretations presented on account of the potential difficulties associated with discerning errors in so new a field. Nevertheless, the literature on the beta anomaly in developing countries is relatively sparse when compared to developed countries, and at times provides drastically different risk–return relationships.

In addition to weaker positive relations than those implied by the CAPM, along with the explanations provided by behavioral finance, negative risk–return relations have been found in several tests of the CAPM (Haugen and Heins 1975; Ang et al. 2006; Frazzini and Pedersen 2014). The implication for negatively sloped SMLS is that high-beta stocks are associated with lower risk-adjusted returns, providing evidence for the beta anomaly. Han et al. (2020) and Hanauer and Lauterbach (2019) have reported a negative relationship for China and emerging markets, in general, respectively. On the JSE, Van Rensburg and Robertson (2003) and Strugnell et al. (2011), have also found a negative relationship between returns and beta. On the other hand, Ward and Muller (2012) cast doubt on the empirical validity of their results, on account of their inclusion of a short time frame and application of a thin trading filter, which, they assert, strongly biased their results towards start-up stocks. Nonetheless, they found, and present with greater conviction, a negative relationship between beta and returns when applying four different test methodologies (Ward and Muller 2012).

On the other hand, negative SML results are met with contention. Karp and Van Vuuren (2019) assert that the CAPM and other asset-pricing models perform poorly on the JSE due to poor proxies. Consequently, the beta anomaly may not be adequately tested for because of the inherent methodological limitations of applying the model in developing countries. Baker and Haugen (2012) provide supporting evidence in the literature and

assert that there are significant differences in beta values between emerging and developed markets, which contribute to the inaccuracy of developing-market betas. The reasons for the significant differences in the literature include the difficulties posed by the inherent heterogeneity of emerging markets, their greater information asymmetries (Lim and Brooks 2010) and low liquidity (Nishiotis 2002), and structural differences in terms of their micro and macro foundations, which have contributed to difficulties in attaining reliable betas (Bekaert and Harvey 1997). Ultimately, the literature on the beta anomaly generally acknowledges its existence, and variable explanations for the anomaly are provided in the literature without a single overarching perspective.

## 3. Data and Methodology

### 3.1. Data

This study uses monthly adjusted closing prices (i.e., adjusted for dividends) for 250 companies listed on the Johannesburg Stock Exchange (JSE), covering a period of 65 months from May 2016 to June 2021. However, some companies have been omitted due to availability constraints, with the minimum number of companies in the study equaling 220. The FTSE/JSE top 40 index is used as a proxy for the market, as it comprises 80% of total market capitalization. The 3-month Treasury bill (T-bill) is used as a proxy for the risk-free rate, as it has been observed to have the lowest market and inflation risk among Treasury bills, particularly over different time periods (Mukherji 2011). It was also selected for its common use in the literature. The factor variables were sourced from Peresec, a South African financial services agency, which provides factor variables constructed according to modern international factor-model specifications introduced by Fama and French. Asset returns employed were computed using log-first differencing on the reported monthly data. The stock prices used were sourced from the Bloomberg database, while the T-bill rate data were sourced from the South African Reserve Bank.

### 3.2. Methodology

The methodological approach in this study is divided into two sections. The first follows the CAPM test employed by Black et al. (1972), which consists of first running a time-series (first-pass) regression to estimate betas and a cross-sectional regression (second pass) to test the CAPM implications. The second section, the robustness check, employs a Fama–French 3 factor model to cross-check the results obtained. The benefits of the methodological approach employed are discussed in the section that follows.

#### 3.2.1. First-Pass Regression

First, individual stock betas were estimated by regressing monthly excess returns against market excess returns, between the months of May 2016 and June 2021, according to Equation (1), below:

$$r_{i,s} = r_{f,s} + \beta_{i,s}^{mkt}\left[r_m - r_f\right] + e_{j,s} \tag{1}$$

where $r_i$ is the expected return of the portfolio, $i$; $r_f$ is the risk-free rate; $r_m$ is the expected return on the market portfolio; $\left[r_m - R_f\right]$ is the market risk premium; and $e_{j,s}$ is the error term associated with estimating the expected returns (Sharpe 1964). $\beta_i^{mkt}$ is a measure of the sensitivity of security $i$ to the excess return on the market, and can be expressed as:

$$\beta_i^{mkt} = \frac{Cov(r_i, r_m)}{Var(r_m)} \tag{2}$$

where $Cov(r_i, r_m)$ is the covariance of the security with the return on the market, and $var(r_m)$ is the variance of the return on the market portfolio.

Equation (1) above can also be written as follows:

$$E(R_i) = E(R_m)\beta_i^{mkt} \tag{3}$$

where $(R_i)$ is $\left[r_i - r_f\right]$ and $(R_m)$ is $\left[r_m - r_f\right]$. Equation (3) implies in that $a_i$ may be defined as

$$a_i = E(R_i) - E(R_m)\beta_i^{mkt} \tag{4}$$

From (4), $a_i$ is presumed to be equal to zero, according to the null hypothesis, according to which the market is competitive and efficient without any possibility of arbitrage (Pennacchi 2008). Provided this is the case, there is a positive association between beta and return, based on the positively sloped SML in the CAPM. Alternatively, a value of $a_i$ significantly different from zero warrants the rejection of the null and the conclusion that the market is inefficient (Pennacchi 2008).

In the second step, the 220 stocks are divided into 10 portfolios sorted from highest to lowest beta. Hence, the first portfolio is made up of the 22 highest beta stocks, and the second portfolio consists of the next highest beta stocks, and so on, until the last portfolio of 22 stocks with the lowest betas. The sorting method employed is the same as that employed by Ali and Badhani (2021), with portfolio returns calculated using equal weights for the individual stock returns. The purpose of sorting is to compare portfolio performances with individual returns and beta portfolios with other beta portfolios across different values of betas. Furthermore, equally weighted market capitalization portfolios are constructed for comparison with the beta portfolios. They are sorted, as in the beta portfolios, in descending order of magnitude. However, they are sorted according to market capitalization. Hence, the first portfolios consist of the 22 largest stocks based on market capitalization, and so forth, until the last portfolio of 22 stocks with the lowest market capitalization. Average returns are then computed for the individual assets and portfolios constructed.

### 3.2.2. Second-Pass Regression

Black et al. (1972) argue that the second-pass cross-sectional test supplements the traditional CAPM test defined by Equation (3) by allowing tests of the linearity of the risk–return relationship without having to specify the intercepts $\gamma_0 = 0$ or $\gamma_1 = r_m - r_f$. In the third step (second-pass regression), the cross-sectional model is estimated for the beta- and market-capitalization-sorted portfolio according to the equation below.

$$\overline{R}_i = \gamma_0 + \gamma_1 \hat{\beta}_i + \delta_i \tag{5}$$

where $\overline{R}_i$ is the average return for each portfolio, $i$; $\gamma_0$ is the intercept; $\gamma_1$ is the estimated market risk premium; $\hat{\beta}_i$ is the estimated beta of each portfolio; and $\delta_i$ is the error term associated with the estimation (Pennacchi 2008).

Equation (5) represents the positively sloped SML, and its parameters, $\gamma_0$ and $\gamma_1$, are used to test the CAPM. The CAPM does not hold for the investigated market if $\gamma_0$ is significantly different from 0 and $\gamma_1$ is also significantly different from $r_m - r_f$. Rejecting the null hypothesis then leaves room for the possibility of either a steep or negatively sloped SML, providing strong evidence for the presence of the beta anomaly.

The use of cross-sectional regressions to test for the beta anomaly is accompanied by the possibility that they may introduce biases and measurement errors that counteract the benefits of employing the second-pass regression. In this case, the estimation error associated with the beta estimates may bias the OLS estimates for the parameters $\gamma_0$ and $\gamma_1$ in Equation (5), providing misleading results for tests on the differences between $\hat{y}_0$ and $\hat{y}_1$ (Black et al. 1972).

To deal with these issues, this study has attempted to:

(1) Estimate the betas for the individual assets and adjust them according to the Blume beta adjustment process; raw betas present estimation biases (Blume 1975).

(2) Group the betas into portfolios and use the portfolio mean returns as they provide improved beta estimates (Abdymomunov and Morley 2011; Blume 1975).

(3) Contrast the results of the cross-sorting methods of beta and market capitalization to increase the robustness of the results obtained.

Employing the above significantly reduces the chances of measurement errors and biases, which may provide misleading results, as noted by Black et al. (1972) and Ali and Badhani (2021).

### 3.2.3. Robustness Checks

In consideration of the shorter time frame used in this study, the abnormal levels of uncertainty over the research period, and the widely accepted idea that market beta is unlikely to be the only relevant factor in explaining returns, the Fama–French three-factor model is used to check the robustness[1] of the results obtained (Fama and French 2004). The aim is to observe any variation in the results obtained due to extraction of IVOL from the CAPM, through a change in model specification. From this, a conclusion as to the existence or non-existence of the beta anomaly, in addition to its relationship with IVOL, is reached. The Fama–French three-factor model is estimated according to:

$$r_{i,} = r_{f,} + \beta_i^{mkt} RMRF + \beta_i^{size} SMB + \beta_{i,s}^{value} HML + e_i \tag{6}$$

where $r_{i,}$ is the expected return on asset $i$; $r_f$ is the risk-free rate; $\beta_i^{mkt}$ is the sensitivity of asset $i$ to the market risk premium; $\beta_i^{size}$ is the sensitivity of asset $i$ to $SMH$; and $\beta_{i,s}^{value}$ is the sensitivity of asset $i$ to $HML$ (Fama and French 2004); and $SMB$ (difference between small-market-value-size portfolios and average of large-market-value-size portfolios) and $HML$ (difference between high book-to-market ratio stocks and low book-to-market ratio stocks) are the return on the size and book-to-market replicating portfolios, respectively (Fama and French 2004). The estimation and portfolio formation follow the guidelines set out by Fama and French (2004).

### 4. Descriptive Statistics and Preliminary Analysis

Table 1, below, presents the summary statistics for the beta, the cross-section standard deviation of the returns, the expected time-series return, and the idiosyncratic volatility (IVOL) of 220 stocks on the JSE from May 2016 to June 2021.The betas were estimated through the regression of the excess return on each individual asset to the market excess return, and the IVOL is the standard deviation of the residual terms for the equations used to estimate the beta.

**Table 1.** Summary statistics for estimated standard deviation (SD), beta, average returns, and idiosyncratic volatility (IVOL).

| Descriptor | SD | Beta | IVOL | Return |
|---|---|---|---|---|
| Mean | 0.12 | 0.78 | 0.34 | −0.06 |
| Median | 0.10 | 0.75 | 0.28 | −0.06 |
| Mode | 0.07 | 1.00 | 0.18 | −0.08 |
| SD | 0.08 | 0.54 | 0.24 | 0.02 |
| Kurtosis | 6.81 | 3.94 | 6.79 | 3.46 |
| Skewness | 2.26 | 0.56 | 2.35 | 0.18 |
| Minimum | −0.05 | −1.10 | 0.02 | −0.13 |
| Maximum | 0.57 | 3.11 | 1.58 | 0.00 |
| Count | 228.00 | 228.00 | 228.00 | 228.00 |

Note: Beta–IVOL correlation = 0.153. IVOL–alpha correlation = −0.201.

Table 1 suggests that the mean monthly average excess return is −0.06% per month, which is equivalent to −0.72% per annum. The maximum return is 0.0%, which rounds up to a very small value, but is nonetheless noticeably low. The negative returns could be a reflection of the negative impact of COVID-19 on the JSE. The average estimated beta is 0.78, the idiosyncratic volatility 0.34%, and the mean standard deviation 0.12%. The IVOL and SD differ noticeably in magnitude from those observed for the emerging market of India (Ali and Badhani 2021). The skewness of the returns is 0.18, while the kurtosis value is 3.46 (excess kurtosis of 0.46). This would suggest that the data are characterized

by a mild level of leptokurtosis, and that they are thus very close to normality. Finally, a positive beta–IVOL correlation of 0.153 and a negative IVOL–Alpha correlation of 0.201 can be observed.

Theoretically, the observed normal distribution of returns would suggest that the mean-variance portfolio framework should remain intact in SA (Bekaert et al. 1998). However, these return distributions contradict the empirically observed non-normal returns for emerging markets (Bekaert et al. 1998). Nevertheless, the fatter and longer tails observed from the excess kurtosis are consistent with the empirical data on the stock returns for BRICS (Brazil, Russia, India, China, and South Africa) (Adu et al. 2015). Sorting and investigating the properties of the assets in portfolio form provides further insights into the properties of the variables studied.

The ten portfolios sorted by beta, standard deviation, and IVOL are presented, along with their respective average returns, in Table 2. In the following Figures 1–3, the (b) elements present a visualization of Table 2; therefore, a discussion of the graph covers the content in Table 2.

**Table 2.** Average portfolio returns sorted by standard deviation, beta, and IVOL.

| Decile | Portfolio Sorted by SD | | Portfolio Sorted by Beta | | Portfolio Sorted by IVOL | |
|---|---|---|---|---|---|---|
| | SD | Return (%) | Beta | Return | IVOL | Return |
| 1 | 1.71 | 0.04 | 1.83 | 0.07 | 4.57 | −0.05 |
| 2 | 0.93 | 0.06 | 1.22 | 0.06 | 2.38 | −0.05 |
| 3 | 0.70 | 0.06 | 1.04 | 0.05 | 1.86 | −0.06 |
| 4 | 0.61 | 0.06 | 0.92 | 0.07 | 1.59 | −0.06 |
| 5 | 0.54 | 0.06 | 0.84 | 0.06 | 1.41 | −0.07 |
| 6 | 0.48 | 0.06 | 0.71 | 0.06 | 1.28 | −0.06 |
| 7 | 0.42 | 0.06 | 0.61 | 0.06 | 1.26 | −0.07 |
| 8 | 0.38 | 0.06 | 0.51 | 0.06 | 1.00 | −0.06 |
| 9 | 0.34 | 0.07 | 0.40 | 0.07 | 1.03 | −0.06 |
| 10 | 0.26 | 0.07 | 0.17 | 0.06 | 0.68 | −0.07 |

Note: Table presents decile of stocks sorted in descending order according to standard deviation (SD), beta, and IVOL, respectively. Column 2 represents the SD-sorted portfolio decile, while column 3 represents the returns for each decile. Similarly, the beta- and IVOL-sorted deciles are in columns 4 and 6, while columns 5 and 7 present their returns, respectively.

Figure 1a, below, shows the estimated betas against the average excess returns for the 220 stocks during the period of May 2016 to June 2021. The regression line is downward-sloping (negative SML). This shows that, on a risk-adjusted basis, high-beta stocks are associated with lower returns. These result are counterintuitive, given the expected risk–return relationship implied by the CAPM. However, they are similar to those presented by Jylhä (2018) for US stocks in periods of high (77–100%) initial margin requirements. This is consistent with findings by Han et al. (2020) and Ali and Badhani (2021), who have observed a negatively sloped SML for China and India, respectively. Other studies have observed the negative SML for JSE, which is consistent with expectations for emerging markets, implying that the beta anomaly is present on the JSE (Van Rensburg and Robertson 2003; Strugnell et al. 2011).

Figure 1b represents the relationship between the beta-sorted portfolios and their expected return. The beta-sorted portfolios show a positive risk–return relationship but with a noticeably flatter line. Theoretically, this implies that risk is rewarded, but at a smaller rate than additional risk taken. These results are consistent with those presented by Frazzini and Pedersen (2014) on US assets, where they conclude that the flatness of the SML is not exclusive to US stocks but is rather observed throughout the world. The conflicting individual and portfolio returns suggest that a test for the beta anomaly may provide inconclusive results. However, the flatter slope of the SML in the portfolios suggests a weaker risk–return relationship than expected. This flatter risk–return relationship is also observed for total risk in the data.

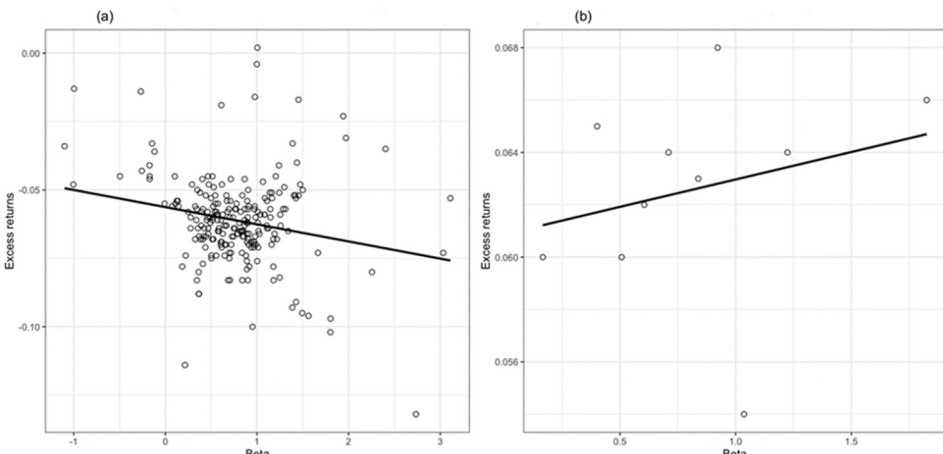

**Figure 1.** Relationship between betas and excess returns for individual stocks (**a**) and portfolio of stocks (**b**).

Figure 2a represents an upward-sloping relation between standard deviation and expected returns of individual stocks. Many of the values are gathered around 0.1, and the slope of the regression is close to flat, implying that investors can expect little reward for taking on a large amount of additional risk. Figure 2b represents the relationship between standard-deviation-sorted portfolios and excess returns. The regression line is downward-sloping with a slope of close to one.

On one hand, the upward-sloping risk–return relationship for individual portfolios (Figure 2a) confirms the significance of total risk in emerging markets, as discussed by Estrada and Serra (2005), albeit without as strong an effect, given the flatness of the line, and different in sign from that observed by Ali and Badhani (2021) for individual stocks. On the other hand, negative slope for the portfolio returns (Figure 2b) is very steep, and significantly steeper than that observed for India, which seems to be a consequence of an outlier in the data, with very low returns for high levels of volatility (Ali and Badhani 2021). The relationship between risk and return would evidently remain negative, but flatter, with the exclusion of the outlier portfolio; however the relevant contradictory implications of the risk–return relationship imply that the theory remains intact.

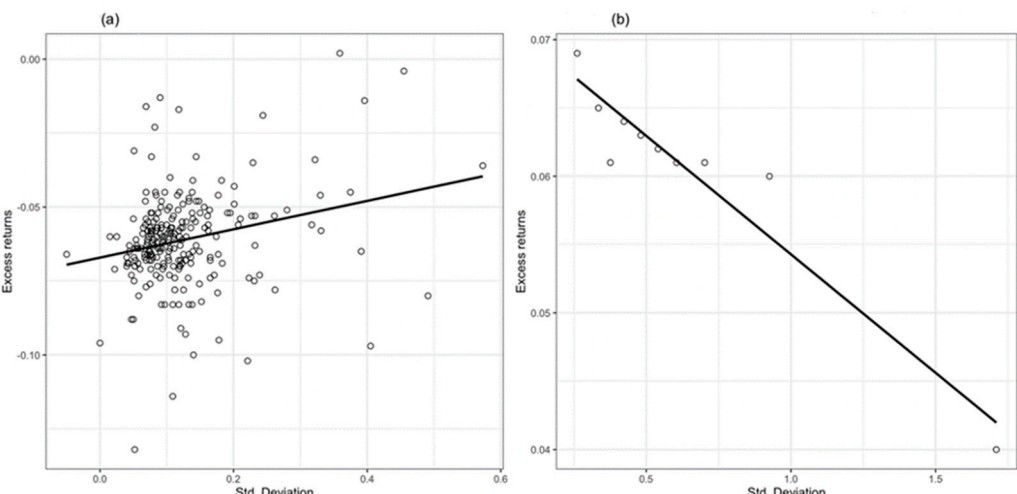

**Figure 2.** Relationship between standard deviation and excess returns for individual stocks (**a**) and portfolio of stocks (**b**).

Figure 3a, above, represents the relationship between the IVOL and the mean excess returns for individual assets. The graph shows a positive and flatter slope between IVOL

and return, suggesting that an investor can expect small returns from huge amounts of additional risk. Similarly, Figure 3b presents a positive relationship for IVOL-sorted portfolios. These results are in stark contrast to those by Stambaugh et al. (2015) and Ang et al. (2006). They also contradict the IVOL–return relationship observed for India (Ali and Badhani 2021). These results run counter to the view that the beta anomaly may be explained according to IVOL, alpha, and beta relations (Liu et al. 2018). The results obtained are still consistent with the theory on the risk–return relationship. Noticeably, average returns for the IVOL-sorted portfolios are negative. The implication is that IVOL-sorted portfolios would not be attractive even though they present a positive risk–return relationship.

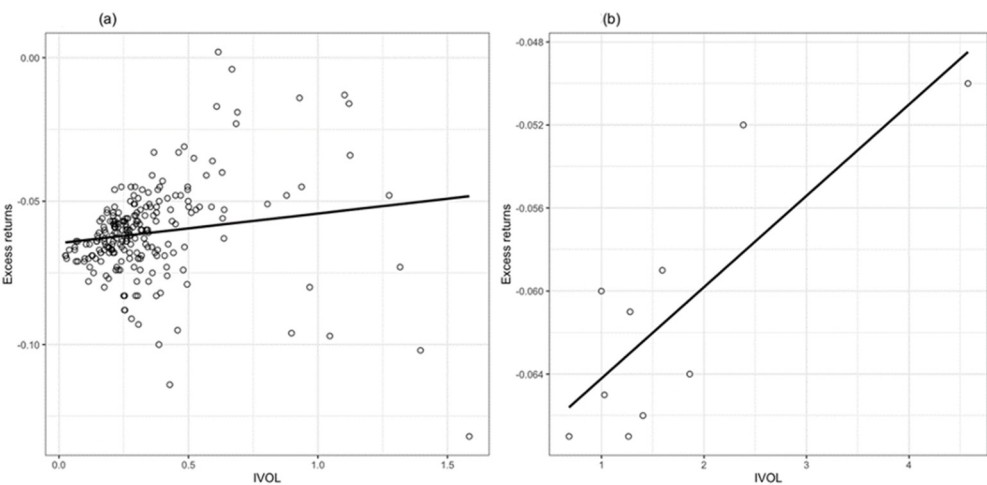

**Figure 3.** Relationship between IVOL and excess returns for individual stocks and portfolio of stocks.

The preliminary analysis presented above provides evidence for the possible presence of the beta anomaly in the JSE. Assets with higher risk, as defined by beta and standard deviation, provide lower returns than assets with lower risk, for individual stocks, on a risk-adjusted basis. Conversely, beta- and standard-deviation-sorted portfolios show positive returns, albeit with a noticeably flat slope, a result that provides reasonable evidence that a weaker risk–return relationship is implied by the CAPM (Frazzini and Pedersen 2014). From this, it is evident that the beta and standard deviation risk–return relationships are inconclusive and contradict the CAPM. On the other hand, assets with high IVOL show positive returns for both individual assets and portfolios. This positive IVOL–return relationship does not seem to offer evidence for the beta anomaly, as indicated in the literature on IVOL–beta relationships (Liu et al. 2018). While the preliminary analysis provides slight evidence for the beta anomaly on the JSE, a more in-depth analysis of this is presented in the next section.

## 5. Results and Discussion

### 5.1. First Pass Regression

Table 3a,b, below, show the summary statistics for the market capitalization (CAP)-sorted decile portfolio estimates and beta-sorted decile portfolios during the investigation period. OLS estimates were used to construct the parameters for the alpha and betas for the 10 portfolios (market-capitalization- and beta-sorted) using monthly data for five years (220 observations). The portfolios (columns) numbered 1 to 10 represent the portfolios sorted in descending order according to market capitalization for Table 3a and the portfolios sorted in descending order according to beta for portfolio 3b. The estimated beta coefficients range from 1.075 to 0.535 (1.87 for portfolio 1 to 0.165 for portfolio 10 for the beta-sorted portfolios). The second row of the tables represents the estimated beta of each portfolio, while the alpha ($a_i$) values are given by the third row, with the t-values directly below them. Lastly, the correlation between the portfolio returns and the return on the market is

given by $corr(r_i, r_m)$ in row 5 and the average monthly excess returns, and their standard deviations, are presented in rows 6 and 7, respectively.

**Table 3.** (**a**) Results for time-series first-pass regression (weighted portfolios) (May 2016 to June 2021); (**b**) results for time-series first-pass regression (beta-sorted portfolios) (May 2016 to June 2021).

| (a) | | | | | | | | | |
|---|---|---|---|---|---|---|---|---|---|
| **(Regression Sample Size 220)** | | | | | | | | | |
| Portfolio | 1 | 2 | 3 | 4 | 5 | 6 | 7 | 8 | 9 | 10 |
| $\hat{\beta}_i$ | 1.075 | 0.883 | 0.954 | 0.745 | 0.818 | 0.813 | 0.535 | 0.806 | 0.536 | 0.733 |
| $a_i$ | 0.005 | −0.008 | 0.001 | −0.019 | −0.009 | −0.005 | −0.039 | −0.009 | −0.041 | −0.013 |
| $t$-stat | −0.269 | −0.711 | −0.432 | −1.312 | −0.480 | −1.009 | −2.165 | −0.819 | −1.880 | −1.064 |
| $corr(r_i, r_m)$ | 0.753 | 0.680 | 0.712 | 0.634 | 0.515 | 0.634 | 0.377 | 0.685 | 0.325 | 0.564 |
| $R_i,$ | −0.007 | 0.000 | 0.002 | 0.003 | 0.006 | 0.009 | 0.005 | 0.008 | 0.003 | 0.012 |
| $R_i, STDEV$ | 0.066 | 0.056 | 0.0608 | 0.0457 | 0.066 | 0.056 | 0.044 | 0.049 | 0.044 | 0.057 |

| (b) | | | | | | | | | |
|---|---|---|---|---|---|---|---|---|---|
| **(Regression Sample Size 220)** | | | | | | | | | |
| Portfolio | 1 | 2 | 3 | 4 | 5 | 6 | 7 | 8 | 9 | 10 |
| $\hat{\beta}_i$ | 1.826 | 1.224 | 1.036 | 0.922 | 0.835 | 0.710 | 0.605 | 0.507 | 0.401 | 0.166 |
| $a_i$ | −0.002 | −0.001 | −0.072 | −0.003 | −0.011 | −0.009 | −0.007 | −0.012 | 0.006 | −0.004 |
| $t$-stat | −0.269 | −0.905 | −0.670 | −1.600 | −0.480 | −1.009 | −1.800 | −0.819 | −1.220 | −0.870 |
| $corr(r_i, r_m)$ | 0.489 | 0.090 | 0.026 | −0.262 | 0.217 | −0.017 | −0.107 | −0.308 | 0.269 | 0.199 |
| $R_i,$ | −0.064 | −0.062 | −0.048 | −0.069 | −0.063 | −0.057 | −0.055 | −0.060 | −0.066 | −0.060 |
| $R_i, STDEV$ | 0.030 | 0.013 | 0.032 | 0.011 | 0.008 | 0.030 | 0.030 | 0.009 | 0.011 | 0.017 |

$R_i,$ is the average monthly excess returns; $R_i, STDEV$ is the standard deviation of excess returns.

According to Table 3a, at the alpha intercepts (row 1), only the first and third portfolio provide positive risk-adjusted returns, while the other portfolios provide negative risk-adjusted returns. Evidently, high- (beta less than 1) and low-risk portfolios (beta greater than 1) provide returns that are marginally consistent with the CAPM but, for the most, part unintuitive. That is, the returns are mostly negative irrespective of the risk level, which suggests that there is no discernible relationship between risk and return. Harvey (1995) has asserted that relatively low and insignificant betas are common for emerging markets; however, Karp and Van Vuuren (2019) have also suggested that bias and noise may lead to inconclusive results, particularity when the sample size and horizon are small. Abnormal events, such as COVID-19 and SA credit downgrading, have likely contributed to the noise in the period studied; therefore, the former explanation is likely. The fact that the absolute values of the returns oscillate between high and low values without a clear direction may also be evidence of the volatility induced by COVID-19. The positive risk-adjusted returns portfolios (column 1 and column 3, Table 3a) have the highest correlation coefficients among all the other items.

The results observed are inconsistent with those observed by Black et al. (1972) for the NYSE and Karp and Van Vuuren (2019) for the JSE. However, the "*t*" values for the alphas in the second column show that only the seventh and ninth portfolios had "*t*" values greater than 1.85, suggesting that the overall results obtained from the first-pass regression may be inconclusive. In their paper, Black et al. (1972) suggested that their results, which showed returns that diverged from the traditional CAPM, may have vastly understated the departures of returns from the model due to some nonstationary presence in the model variables. In anticipation of this issue, this paper made use of log-first differenced variables to reduce the effects of non-stationarity. However, despite this, the results obtained for both types of portfolio in the first-pass time-series regression did not lead to definitive conclusions about the beta anomaly on the JSE. The second-pass cross-sectional regression was then used to test for the anomaly further.

### 5.2. Second Pass Empirical Results

Table 4a,b show the summary statistics for the second-pass regressions for the market-capitalization-sorted portfolios (Figure 4a) and beta-sorted portfolios (Figure 4b) over the investigation period for 65 observations and 220 assets. The statistics and OLS estimates ($\hat{y}_0$ and $\hat{y}_1$) were obtained by regressing the average excess returns of the portfolios against their estimated betas, according to Equation (5). The betas used for the cross-sectional regression were the adjusted betas obtained from the time-series first-pass regressions. The portfolios (columns) numbered 1 to 10 represent portfolios sorted in descending order according to market capitalization (Figure 4a) and portfolios sorted in descending order according to the betas (Figure 4b). For each table, the first and second row provide OLS estimates for the parameters $\gamma_0$ and $\gamma_1$, respectively. The *t*-stats for the parameters are provided in row 3 and row 4, while the standard errors for the parameters are provided in rows 5 and 6, respectively. Finally, the R-squared values are provided in row 7. The theoretical slope, $r_m - r_f$, is equal to $-0.0691953$.

**Table 4.** (**a**) Results for cross-sectional second-pass regression (weighted portfolios) (May 2016 to June 2021); (**b**) results for cross-sectional second-pass regression (beta portfolios) (May 2016 to June 2021).

| (a) | | | | | | | | | | |
|---|---|---|---|---|---|---|---|---|---|---|
| **(Regression Sample Size 220)** | | | | | | | | | | |
| *Port* | 1 | 2 | 3 | 4 | 5 | 6 | 7 | 8 | 9 | 10 |
| $\hat{y}_0$ | −0.074 | −0.063 | −0.077 | −0.075 | −0.068 | −0.062 | −0.062 | −0.073 | −0.076 | −0.042 |
| $\hat{y}_1$ | 0.002 | −0.002 | 0.014 | 0.018 | 0.010 | 0.006 | 0.004 | 0.018 | 0.026 | −0.018 |
| *t*-stat ($\hat{y}_0$) | −12.047 | −9.315 | −13.556 | −11.712 | −12.014 | −10.726 | −13.049 | −11.525 | −11.652 | −6.296 |
| *t*-stat ($\hat{y}_1$) | 0.344 | −0.285 | 2.502 | 2.161 | 1.619 | 1.048 | 0.534 | 2.753 | 2.544 | −3.450 |
| se ($\hat{y}_0$) | 0.006 | 0.007 | 0.006 | 0.006 | 0.006 | 0.006 | 0.005 | 0.006 | 0.006 | 0.007 |
| Se ($\hat{y}_1$) | 0.005 | 0.007 | 0.006 | 0.008 | 0.006 | 0.006 | 0.008 | 0.007 | 0.010 | 0.005 |
| R squared | −0.044 | −0.046 | 0.200 | 0.149 | 0.072 | 0.005 | −0.035 | 0.239 | 0.207 | 0.342 |
| (b) | | | | | | | | | | |
| **(Regression Sample Size 220)** | | | | | | | | | | |
| *Port* | 1 | 2 | 3 | 4 | 5 | 6 | 7 | 8 | 9 | 10 |
| $\hat{y}_0$ | −0.039 | −0.012 | −0.083 | −0.014 | −0.042 | 0.150 | −0.251 | −0.044 | −0.104 | −0.050 |
| $\hat{y}_1$ | −0.013 | −0.041 | 0.034 | −0.060 | −0.025 | −0.292 | 0.324 | −0.032 | 0.094 | −0.056 |
| *t*-stat ($\hat{y}_0$) | −1.705 | −0.233 | −0.083 | −0.178 | −1.014 | 1.069 | −1.967 | −1.211 | −3.672 | −10.938 |
| *t*-stat ($\hat{y}_1$) | −1.083 | −0.974 | 0.034 | −0.719 | −0.515 | −1.476 | 1.538 | −0.451 | 1.339 | −2.776 |
| se ($\hat{y}_0$) | 0.023 | 0.052 | 0.147 | 0.077 | 0.041 | 0.141 | 0.128 | 0.036 | 0.028 | 0.005 |
| Se ($\hat{y}_1$) | 0.012 | 0.042 | 0.142 | 0.083 | 0.049 | 0.198 | 0.211 | 0.071 | 0.070 | 0.020 |
| R squared | 0.055 | 0.045 | 0.003 | 0.025 | 0.013 | 0.098 | 0.106 | 0.010 | 0.082 | 0.278 |

Note: $\hat{y}_0$ *p*-values all significant at 1% level.

The formal test of the CAPM was performed via the hypothesis test conducted on the parameters from Equation (5), as follows:

$$H_0 : \gamma_0 = 0$$

$$H_1 : \gamma_0 \neq 0$$

$$H_0 : \gamma_1 = r_m - r_f = -0.0691953$$

$$H_1 : \gamma_1 \neq r_m - r_f = -0.0691953$$

where Equation (5) represents the security market line and the slope, $\gamma_1$, is an estimate of the market risk premium. Thus, if the CAPM holds, $\gamma_0$ should be equal to zero and $\gamma_1$ equal to the market portfolio means excess returns, $r_m - r_f$. The test conducted is a two-tail test with a 95% confidence interval and critical value of 1.960. Given the alternative hypothesis of $\gamma_0 \neq 0$, the mean value of $\gamma_0$ could either be on the left or the right side of $\gamma_0$; hence, a two-tail test was used.

In Table 4a, the empirical slopes are flat, with values close to zero. They increase slightly as market capitalization decreases from portfolio 1 to 10, with the final value being negative. The empirical slopes are also less than the theoretical slope, on average, which was also observed by Black et al. (1972). For the beta portfolios (Table 4a), the *t*-stats for $\gamma_0$ are all larger than the critical value of 1.960, which provides enough evidence to reject the null hypothesis that $\gamma_0 = 0$ with a 95% level of confidence. The p-values for the intercept are also all significant at the 1% level. Moreover, the same conclusion may be reached for $\gamma_0$ for the first, seventh, ninth, and tenth portfolio on the beta-sorted portfolios; however, the ninth portfolio is the only one with a statistically significant *p*-value below the 5% level. These results show that, for most of the portfolios investigated, $\gamma_0$ is significantly different from zero, and one of the conditions for the CAPM to hold is not met.

The *t*-stats for the $\gamma_1$ in the weighted portfolios are larger than the critical value for the third, fourth, and eighth to tenth portfolios, which suggests that we may reject the null hypothesis that $\gamma_1$ is equal to the market portfolio mean excess returns. The p-values indicate significance at the 10% level for the weighted portfolios, presenting weak evidence for the conclusion that $\gamma_1$ is significantly different from $r_m - r_f$. However, the R-squared value for the portfolios is above 0.20, indicating that at least 20% of the variation in the portfolio securities' returns may be explained by the beta. For the beta-sorted portfolios, only the final portfolio has a *t*-stat greater than the critical value. The results from the market-capitalization- and beta-sorted portfolios suggest that $\gamma_1$ is significantly different from $r_m - r_f$ as market capitalization and beta decreases, which is a surprising finding.

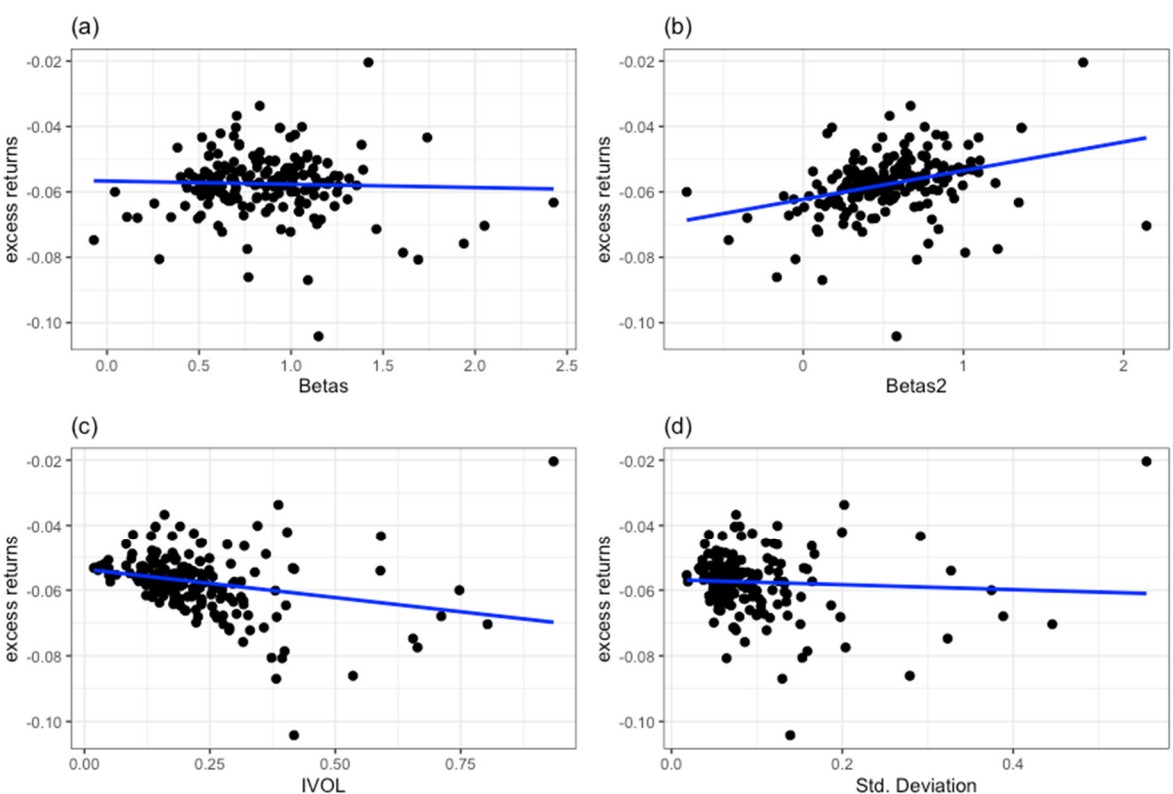

**Figure 4.** Relationship between beta, IVOL, standard deviation, and individual stock excess returns.

### 5.3. Implications for the Beta Anomaly

The empirical evidence provided does not corroborate the positive risk–return relationship implied by the CAPM. First, the slope of the SML is flatter than that implied by the CAPM, which is a similar result to that initially presented by Black et al. (1972). The risk–return relationship is not only weaker than anticipated, but is closer to zero than expected, suggesting the insignificant beta–return relationship presented in the literature

(Harvey 1995). Secondly, a negative risk–return relationship was also observed for a significant number of portfolio returns, a result similar to those observed by several researchers (Haugen and Heins 1975; Ang et al. 2006; and Frazzini and Pedersen 2014). This was the case for some of the market-capitalization- and beta-sorted portfolios. The negative beta–return relationships obtained from the results are comparable to others observed on the JSE (Van Rensburg and Robertson 2003; Strugnell et al. 2011). According to these results, it is likely that the beta anomaly does hold for the JSE.

On the other hand, the results obtained do not present unequivocal evidence for the beta anomaly. This may be indicative of the contention that, due to poor proxies, the CAPM and, subsequently, the beta anomaly, cannot be calculated for in the JSE (Karp and Van Vuuren 2019). Similarly, the alternative view that methodological limitations on the applicability of asset-pricing models to emerging markets, as a result of their heterogenic and idiosyncratic factors, along with their non-normal distributions, may provide a reasonable justification for the weak results obtained (Bekaert et al. 1998). However, it is plausible that the turbulent period covered by the study, in addition to the shorter time frame, may have contributed to the less definitive nature of the results observed. For this reason, before drawing conclusions as to the beta anomaly and its implications for the JSE, the Fama–French three-factor model was implemented with variable parameters (a longer period of 10 years, albeit with fewer assets, at 186) and noise period controls (COVID-19 and downgrading period) to obtain a better insight into the beta anomaly on the JSE.

*5.4. Factor Model and Robustness Checks*

Figure 4a, below, shows the estimated betas against the average excess returns for 186 stocks on the JSE during the sample period of September 2011 to August 2021, using the CAPM. Evidently, even with a larger time horizon, the SML is still negatively sloped. This provides stronger evidence of the presence of the beta anomaly on the JSE, as previously observed (Karp and Van Vuuren 2019; Van Rensburg and Robertson 2003; and Strugnell et al. 2011). Figure 4b, below, provides insights into the possible explanation for the presence of the anomaly. Figure 4b plots the Fama–French-estimated betas against the mean excess returns for the September 2011 to August 2021 period. From the graph, it is evident that the SML is positively sloped, and that it changed from a previously negative slope. This provides strong evidence for the beta anomaly on the JSE and the view that the beta is not the only factor determining the risk–return relationship (Fama and French 1993; Dowen 1988).

Furthermore, the change in the slope of the SML indicates the elimination of omitted variable bias (OVB) from the CAPM, due to the removal of the SMB and HML from the error term, which were parts of the IVOL in the CAPM, supporting the argument that IVOL can explain the beta anomaly (Liu et al. 2018). Indeed, controlling for IVOL removed the anomaly and restored the risk–return relationship implied by the theory.

Figure 4c, above, shows the estimated IVOL against the average excess returns for 186 stocks on the JSE during the sample period of September 2011 to August 2021. According to the IVOL–return relationship on the graph, high-IVOL portfolios provide lower average excess returns than low-IVOL portfolios. In this instance, unlike the possible IVOL–return relationship in Figure 3a, the negative IVOL–return relationship is likely to be reflective of the JSE environment, given the wider time horizon and control for the effects of the COVID-19 pandemic. Lastly, Figure 4d, above, shows the estimated standard deviation against the average excess returns for 186 stocks on the JSE during the sample period of September 2011 to August 2021. As with the IVOL and beta above, a negative risk–return relationship is observed. Consequently, increasing the time horizon, employing the Fama–French three-factor model, as well as controlling for COVID, demonstrates that the beta anomaly and negative risk–return relationship are present on the JSE.

## 6. Conclusions

This study examined the beta anomaly in the Johannesburg Stock Exchange. It highlighted the importance of adequately identifying and explaining asset-pricing anomalies in

emerging markets, as these markets differ significantly in their return distributions from developed markets. While prospective gains from the diversification in emerging markets may be obtained, identifying the variables that explain the risk–return relationship in these markets remains elusive, especially with respect to the beta anomaly. To test the hypothesis for the beta anomaly, asset data from the JSE spanning 10 years and 220 assets, including Treasury bill and risk factor data, were used. From a research perspective, variable accounts and explanations have been provided to explain the beta anomaly and its presence, albeit without on the emergence of a dominant perspective. Nevertheless, a handful of perspectives, namely the behavioral finance and model mis-specification views of the CAPM, have risen above others in both developed and developing market literature on the beta anomaly.

The findings from this study provide evidence for the existence of the beta anomaly in South Africa. They are similar to recent results on the JSE (Van Rensburg and Robertson 2003; Strugnell et al. 2011). The employed graphical analyses and regressions provide evidence for a negatively sloped SML on the JSE, indicating that the beta is not the only determinant of risk on the South African stock market. These results are further strengthened by the positive beta–IVOL correlations and results from the employed t-tests, which reveal that the CAPM does not hold for the JSE as higher risk-adjusted returns are associated with low-beta, low-IVOL, and low-volatility stocks. The working hypothesis that the beta anomaly is driven by IVOL and positive IVOL–beta relations, particularly in overpriced stocks, is further corroborated by the robustness results from the Fama–French three-factor model employed, in which the beta anomaly disappears when controlling for IVOL and for the adverse effects of COVID-19 with an extended study period (Liu et al. 2018).

Considering the results obtained, it is recommended that institutional and individual investors pursue low-volatility construction strategies for the JSE, as previously suggested by Bradfield and Oladele (2018). Further benefits for investors may be obtained in the form of asset diversification gains from investing in different assets. To promote investment, it is recommended that the domestic government and policy makers pass legislation that (1) facilitates the transference of relevant information to investors; (2) reinforces a system of notification and consultation that permits ease of input from all relevant parties; and (3) establishes a public appeals process that effectively addresses dispute settlements. The aim of the policy is to increase transparency in a manner that is consistent with increasing investor confidence, encouraging investment and reducing the uncertainty over investment in South Africa.

Evaluating the results obtained, several limitations and insights for future studies may be noted. Firstly, the results were obtained during a particularly turbulent period and narrow time frame, which may have affected their accuracy. Secondly, the CAPM prevents analysis in great depth, and the use of the Fama–French five-factor model, or another multifactor model, may produce more fruitful results. Future studies may further benefit from observing the period analyzed in this study along with pre-and post-periods once a certain amount of time and, more specifically, COVID-19, have passed.

Future research on the beta anomaly in South Africa should attempt to explore the potential interactions between the beta anomaly and other mark anomalies, along with the potential role of leverage in explaining these relationships. It would also be interesting to examine the key macro and micro factors that drive the beta anomaly.

**Author Contributions:** Methodology, software, investigation, writing—original draft preparation, M.S. Conceptualization, supervision, writing—review and editing, G.N. All authors have read and agreed to the published version of the manuscript.

**Funding:** Godfrey Ndlovu would like to gratefully acknowledge the funding from the University of Cape Town, research grant number 461186.

**Data Availability Statement:** The data presented in this study are available on request from the corresponding author.

**Acknowledgments:** We would like to acknowledge the three anonymous reviewers for their views and comments.

**Conflicts of Interest:** The authors declare no conflict of interest.

## Note

[1]    The other option would be to use the Carhart four-factor model or the Fama–French five-factor model, or any equivalent model. However, we used the Fama–French three-model due to its versatility; it is also widely used in similar studies, thus making it easy to draw comparisons.

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
