# Peer review of "An Investigation of the Beta Anomaly in Emerging Markets: A South African Case"

_jrfm, doi:10.3390/jrfm15050214_

Round 1

Reviewer 1 Report

Concerns:

  1. High risk with low return is also found in GARCH-M model. Is it the result of beta anomaly? 
  2. Line 275: what is "adjusted" closing price? dividend adjusted?
  3. Line 289: from the section of Methodology, the authors used first pass regression and second pass regression to illustrate the beta anomaly. Is it a new method? 
  4. Line 379: the Authors apply Fama-French three factor model for robutness checks. Why use three factor model? how to justify it? how about four factor or seven factor models which are found in literature? 
  5. Line 431:  The title of Table 2 is sorted portfolio! no other description for the reults in Table 2?
  6. Line 403: It is not easy to understand the title of Table 1.
  7. Line 519 and 525: from Table 3a and b, "Statistic" should be "Statistics"? "Times" should be "Time"? What is "Port"? 
  8. Line 539: from Table 4a, "Statistic" should be "Statistics"? 
  9. Line 570: would it be better to make Table 4b neater?
  10. Line 665: simmarize clearly the contribution of the paper and why the topic is interesting
  11. Line 731: from reference, the literatures are not shown in uniform format. 
  12. Line 731: There are some omitted references. For example, Fama and French (2004) cannot be found and only Fama and French (1993) is found.  

Reviewer 2 Report

The paper is interesting and well written. I have a few comments mostly related to the presentation and formatting

In general, you should more carefully describe the procedure followed to comput the Fama-French factors for your data. I would also eventually report some descriptive statistics for the factors-

Page 8: the text from line 365 to 371 must be aligned to the margins

Page 8: line 372: "chaces" should be "chances"

Table 1: the caption is missing

Page 9, line 416. I would say that the data are characterized by tails that are only slightly heavier than what implied by normality. So I would say that the data are characterized by a mild level of leptokurtosis bringing us close to the normality case.

Adjust the width of the tables in the text to fit the margins of the document

The caption of figure 4a should be partly moved to the main text. In the caption I would leave a summary description of the content while the discussion of results should be embodied in the main text.

Reviewer 3 Report

Title: An investigation of the beta-anomaly in emerging markets: A South African Case.

I appreciate the chance to serve as a reviewer on this paper. The paper is well written and suits the scope of the issue. I believe that the paper could be of interest to an international audience since it deals with a very interesting topic. I recommend accepting the paper after minor revisions, so I urge the author(s) to consider the following comments and improve the paper accordingly.

  1. It would be useful if the authors explain in detail how the market regulators benefit by their results.
  2. It would be useful if the authors explain how their model catch up asymmetries. (see for instance Floros et al. 2020).
  3. What will be the economic implications in macro – level in business activity?
  4. It would also be useful for the audience and future researchers if a guide for the future research is provided: how this research could be used concretely to open new pathways? Is it possible to provide some examples and possible directions for future research?

This is a good work and I think that a revised version with the abovementioned concerns could be a contribution to the literature.

Literature

Floros C., Gkillas K., Konstantatos C., Tsagkanos A., (2020) “Realized measures to explain volatility changes over time” Journal of Risk Financial Management. Vol 13(6), 125-144

Round 2

Reviewer 1 Report

The overall revisions are fine. The responses to my concerns are well. 

Author Response

These have been attended to see attached manuscript with tracked changes